# Estimation of sulfuric acid concentration using ambient ion composition and concentration data obtained by Atmospheric Pressure interface Time-of-Flight ion mass spectrometer

Lisa J. Beck[1], Siegfried Schobesberger[2], Mikko Sipilä[1], Veli-Matti Kerminen[1,4] and Markku Kulmala[1,3,4,5]

[1]Institute for Atmospheric and Earth System Research/Physics, University of Helsinki, 00014 Helsinki, Finland

[2]Department of Applied Physics, University of Eastern Finland, 70211 Kuopio, Finland

[3]Aerosol and Haze Laboratory, Beijing Advanced Innovation Center for Soft Matter Sciences and Engineering, Beijing University of Chemical Technology (BUCT), Beijing, China

[4]Joint International Research Laboratory of Atmospheric and Earth System Sciences, School of Atmospheric Sciences, Nanjing University, Nanjing, China

[5]Faculty of Geography, Lomonosov Moscow State University, Moscow, Russia

*Correspondence to:* Lisa Beck (lisa.beck@helsinki.fi) and Markku Kulmala (markku.kulmala@helsinki.fi)

**Abstract**

Sulfuric acid ($H_2SO_4$, SA) is the key compound in atmospheric new particle formation. Therefore, it is crucial to observe its concentration with sensitive instrumentation, such as chemical ionisation (CI) inlets coupled to Atmospheric Pressure interface Time-of-Flight mass spectrometers (APi-TOF). However, there are environmental conditions and physical reasons when chemical ionisation cannot be used, for example in certain remote places or flight measurements with limitations regarding chemicals. Here, we propose a theoretical method to estimate the SA concentration based on ambient ion composition and concentration measurements that are achieved by APi-TOF alone. We derive a theoretical expression to estimate SA concentration and validate it with accurate CI-APi-TOF observations. Our validation shows that the developed estimate works well during daytime in the boreal forest ($R^2 = 0.85$), however it underestimates the SA concentration in e.g. Antarctic atmosphere during new particle formation events where the dominating pathway for nucleation involves sulfuric acid and a base ($R^2 = 0.48$).

## 1 Introduction

Sulfuric acid ($H_2SO_4$, SA) is the key compound in atmospheric new particle formation (e.g. Weber et al., 1995, 1996; Birmili et al., 2003; Kulmala et al., 2004; Kuang et al., 2008; Kerminen et al., 2010; Wang et al., 2011; Kulmala et al., 2014; Yao et al., 2018; Cai et al., 2021), therefore it is crucial to have accurate observations of its concentration. However, ambient concentrations of $H_2SO_4$ are low, commonly less than a part per trillion by volume (~$2\cdot10^7$ molecules cm$^{-3}$), making it challenging to measure it. During the recent years there have been instrumental developments towards a reliable detection of $H_2SO_4$ in the atmosphere, particularly via the development of a Chemical Ionisation Atmospheric Pressure interface Time-of-Flight mass spectrometer (CI-

APi·TOF, Jokinen et al., 2012), using nitric acid as a reagent ion. Still, the measurement technique with CI-APi-
TOF is relatively challenging, as a thorough calibration i.e. with sulfuric acid as proposed by Kürten et al. (2012),
is needed in order to get reliable numbers. Furthermore, the loss of sulfuric acid to surfaces, such as an inlet, and
the correct flow rates must be known and characterised.

During the past decade, Atmospheric Pressure interface Time-of-Flight mass spectrometers (APi-TOF, Junninen
et al., 2010) have been deployed in several measurement campaigns where the use of a CI inlet was either not
possible or desired. In these instances, the APi-TOF only observed the composition and concentration of ambient
ions. The APi-TOF is capable of directly sampling and detecting naturally charged gas-phase ions, including
molecular clusters, and is often being used to detect clustering processes as a first step of new particle formation
on a molecular basis (e.g. Schobesberger et al., 2013; Jokinen et al., 2018; Beck et al., 2021). While a CI-APi-
TOF at best has a limit of detection of around $\sim 10^4$ molecules $cm^{-3}$ ($\sim$ ppq level), the APi-TOF can detect
approximately 1% of the ambient ion concentration (Fig. 1, Junninen et al., 2010). With an average ion
concentration of $\sim$1000 $cm^{-3}$ per polarity (Hirsikko et al., 2011), the APi-TOF is measuring 10 ions $cm^{-3}s^{-1}$ with a
limit of detection of $\sim$0.01 counts per second, hence 0.1 ions $cm^{-3}$. This corresponds to approximately a pps level
($100 \cdot 10^{-21}$), showing that the limit of detection of an APi-TOF in comparison to a CI-APi-TOF is lower by five
orders of magnitudes.

A detailed description of the APi-TOF can be found in Junninen et al. (2010). Since concentrations of neutral
clusters are below the detection limit of CI-APi-TOF in many atmospheric conditions and environments, using
the APi-TOF is currently the only way to directly detect atmospheric clustering. Therefore, if we can estimate
$H_2SO_4$ concentration particularly during initial steps of new particle formation, based on the same dataset, we can
readily get better insight into the process itself.

Since there are only limited long term observations of $H_2SO_4$ concentrations, several proxies on this concentration
have been developed (e.g. Petäjä et al., 2009; Mikkonen et al., 2011; Lu et al., 2019; Dada et al., 2020). These
proxies attempt to approximate the ambient $H_2SO_4$ concentrations using more readily measured quantities, in
particular the sulfur dioxide concentration, (UV) radiation intensity and pre-existing particle number size
distribution that can be used to calculate the condensation sink for gas-phase $H_2SO_4$. In circumstances where the
required data for $H_2SO_4$ proxies are not available, but measurements with an APi-TOF were conducted, the $H_2SO_4$
concentration can be obtained from the ion mass spectra. A first attempt of estimating the sulfuric acid
concentration via the concentration of atmospheric ions was introduced by Arnold and Fabian (1980), followed
by Eisele (1989) under the assumption that most $H_2SO_4$ molecules are charged by reacting with $NO_3-$.

Motivated by the reasonings outlined above, we derive here an expression to estimate $H_2SO_4$ concentration based
primarily on APi-TOF observations and validate it.



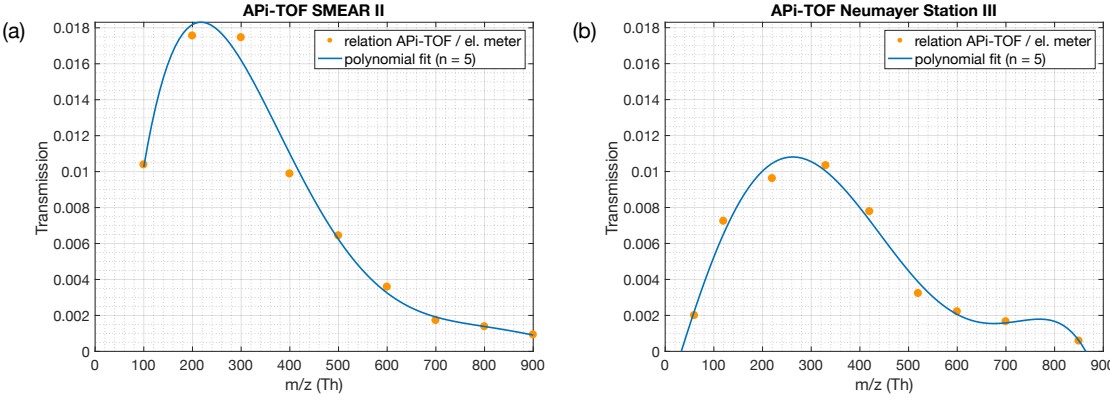

**Figure 1** Ion transmission of the APi-TOFs used in this study. The transmission efficiency was determined via production of charged particles with a NiCr wire. The concentration of the size selected ions with a Hermann nano differential mobility analyser (HDMA, Hermann, 2000) were measured with an electrometer and an APi-TOF in parallel. A more detailed description can be found in Junninen et al. (2010). Panel (a) shows the transmission efficiency of the APi-TOF used for measurements at the SMEAR II Station, Hyytiälä, Finland. Panel (b) shows the transmission efficiency used for measurements at the Neumayer Station III.

## 2 Theoretical estimation of sulfuric acid concentration with bisulphate ion and $H_2SO_4$ clusters

Ambient ion mass spectra have usually clear evidence of gas-phase $H_2SO_4$, predominantly in the form of bisulphate ion ($HSO_4^-$) and its adducts involving $H_2SO_4$, forming so-called dimers ($H_2SO_4 \cdot HSO_4^-$) as well as larger clusters (Ehn et al., 2010). These ions are due to the efficient scavenging of a negative charge by ambient $H_2SO_4$ via proton donation, and due to the high stability of the sulfuric acid-bisulphate ion clusters, in particular for the dimer (Ortega et al., 2014). In order to estimate the sulfuric acid concentration ($H_2SO_4$) using measured naturally charged ions (see Fig. 2), we approximate this concentration by following the bisulphate ion $HSO_4^-$, herein denoted $SA_{monomer}$, the dimer cluster $H_2SO_4 \cdot HSO_4^-$ ($SA_{dimer}$) and trimer cluster $(H_2SO_4)_2 \cdot HSO_4^-$ ($SA_{trimer}$). Any other $H_2SO_4$-containing ion clusters, in particular those larger than the $SA_{trimer}$, typically occur at much smaller concentrations and will be neglected here.

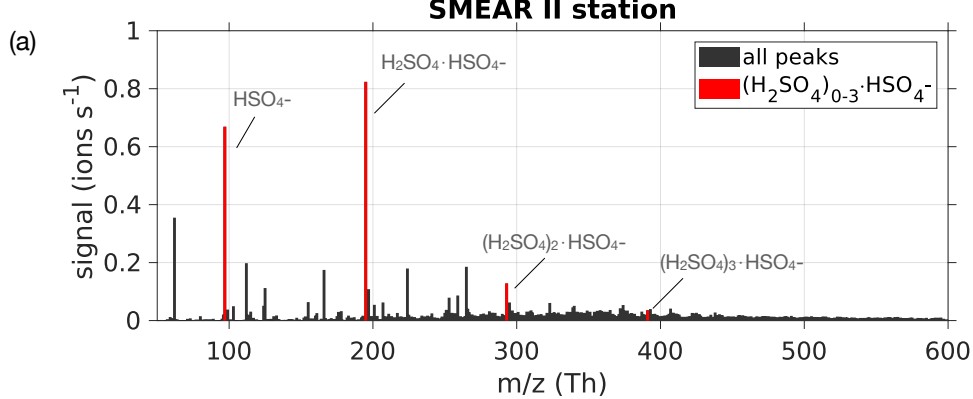

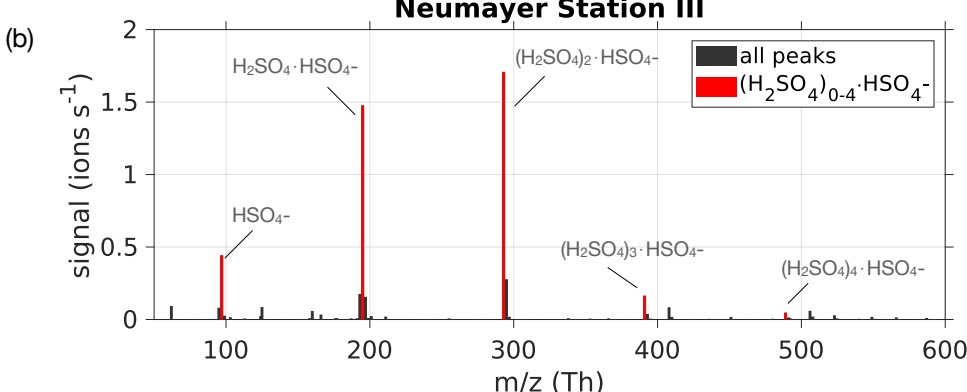

**Figure 2** (a) Mass spectrum from 50 to 600 Th measured with the APi-TOF on 24 May 2017 during the time period 08:00 – 18:00 (local time) at SMEAR II station, Hyytiälä, Finland. (b) Mass spectrum from 14 January 2019 between 08:00 and 18:00 (local time) at Neumayer Station III, Antarctica during a new particle formation event. The bisulphate ion $HSO_4^-$ and $H_2SO_4$ clusters containing it were used for the estimation of $H_2SO_4$ concentration, and are coloured in red.

If we assume that the concentration of $SA_{monomer}$ depends generally on its production rate ($P_1$) and that its loss is by condensation onto aerosol particles (condensation sink, CS), to the $SA_{dimer}$ when clustering with another $H_2SO_4$ molecule, and to ion-ion recombination with positive ions ($N_{pos}$), we get the following equation for the $SA_{monomer}$ concentration:

$$\frac{d[SA_{monomer}]}{dt} = P_1 - CS \cdot [SA_{monomer}] - P_2 - \alpha \cdot [SA_{monomer}] \cdot N_{pos}, \tag{1}$$

where $P_2 = k_1 \times [SA_{monomer}] \times [H_2SO_4]$ is the dimer production rate due to $SA_{monomer}$-$H_2SO_4$ collisions, α ($\approx 1.6 \times 10^{-6}$ cm$^3$ s$^{-1}$) is the ion-ion recombination coefficient (Kontkanen et al., 2013), and the collision rate $k_1$ is assumed to be constant.

For the dimer concentration we consider the production $P_2$, the loss due to CS, the clustering of the $SA_{dimer}$ with
$H_2SO_4$ with a rate constant $k_2$, and the ion-ion recombination:

$$\frac{d[SA_{dimer}]}{dt} = P_2 - CS \cdot [SA_{dimer}] - k_2 \cdot [SA_{dimer}] \cdot [H_2SO_4] - \alpha \cdot [SA_{dimer}] \cdot N_{pos}, \qquad (2)$$


And with substituting $P_2$, eq. 2 for $SA_{dimer}$ changes to:

$$\frac{d[SA_{dimer}]}{dt} = k_1 \cdot [SA_{monomer}] \cdot [H_2SO_4] - CS \cdot [SA_{dimer}] - k_2 \cdot [SA_{dimer}] \cdot [H_2SO_4] - \qquad (3)$$
$$\alpha \cdot [SA_{dimer}] \cdot N_{pos}.$$


Finally, to produce $SA_{trimer}$ we consider the collision of the $SA_{dimer}$ with $H_2SO_4$ and the loss to the CS and ion-ion
recombination. For the sake of completeness, we would additionally have to consider the loss of $SA_{trimers}$ to form
the tetramer $(H_2SO_4)_3 \cdot HSO_4$, however this additional term is rather small and will therefore be neglected in this
derivation. Therefore, we get the simplified equation for $SA_{trimer}$:

$$\frac{d[SA_{trimer}]}{dt} = k_2 \cdot [SA_{dimer}] \cdot [H_2SO_4] - CS \cdot [SA_{trimer}] - \alpha \cdot [SA_{trimer}] \cdot N_{pos}. \qquad (4)$$


For simplification, we consider a pseudo-steady state condition for both dimers and trimers by setting the left-
hand side of eqs. (3) and (4) to be zero, which is justified when the dimer and trimer concentrations change at
rates smaller than their overall production and loss rates. Thereby, from eq. (3) we obtain:

$$k_1 \cdot [SA_{monomer}] \cdot [H_2SO_4] \qquad (5)$$
$$= CS \cdot [SA_{dimer}] + k_2 \cdot [SA_{dimer}] \cdot [H_2SO_4] + \alpha \cdot [SA_{dimer}] \cdot N_{pos}$$


and from eq. (4) we obtain:

$$k_2 \cdot [SA_{dimer}] \cdot [H_2SO_4] = CS \cdot [SA_{trimer}] + \alpha \cdot [SA_{trimer}] \cdot N_{pos}. \qquad (6)$$


If we now deploy equation (6) in equation (5) and solve for $H_2SO_4$, the result is:

$$k_1 \cdot [SA_{monomer}] \cdot [H_2SO_4] = CS \cdot [SA_{dimer}] + CS \cdot [SA_{trimer}] + \alpha \cdot [SA_{dimer}] \cdot \qquad (7)$$
$$N_{pos} + \alpha \cdot [SA_{trimer}] \cdot N_{pos},$$

$$[H_2SO_4] = \frac{(CS + \alpha \cdot N_{pos}) \cdot ([SA_{dimer}] + [SA_{trimer}])}{k_1 \cdot [SA_{monomer}]}. \qquad (8)$$


Besides the steady-state assumption, it should be noted that in deriving eq. 8 monomers, dimers and trimers were
assumed to have the same loss rate (CS) onto pre-existing aerosol particles. This causes an additional, yet minor,
uncertainty in the estimated $H_2SO_4$ concentrations, as such loss rates are dependent on the size/mass of the clusters
(e.g. Lehtinen et al., 2007; Tuovinen et al., 2021). According to Tuovinen et al. (2021), the CS of $H_2SO_4$ clusters
decreases with increasing number of $H_2SO_4$ molecules. The study shows that the CS of the $SA_{dimer}$ clustered with
ammonia decreases to 68% (compared to one $H_2SO_4$ molecule) and for $SA_{pentamer}$ with four ammonia molecules
to 42%. However, the order of magnitude of the CS remains the same, and the effect on the estimation of the
$H_2SO_4$ concentration is assumed to be negligible. Additionally, the CS for ions is higher than for neutral
compounds. The enhancement of CS has shown to reach a maximum value of 2 when the pre-existing particles
are < 10 nm and decreases to 1 when the pre-existing particles are > 100 nm, as shown by Mahfouz and Donahue
(2021). The impact of ions on CS and estimated SA concentrations depends thereby on the environmental
conditions determining the size distribution and charges of the pre-existing particle population. Neglecting the
size-dependency of CS between the SA monomers, dimers and trimers causes additional errors in estimated SA
concentrations; however, it is difficult to determine this effect in ambient measurements having limited data and
instrumentation.

Furthermore, the derivation neglects the losses of $SA_{trimer}$ to the $SA_{tetramer}$ and larger clusters, as well as the
clustering of sulfuric acid ion clusters with water and base molecules, such as $NH_3$. Those simplifications can
cause an underestimation of the $H_2SO_4$ concentration with the presented method. If necessary, the method can
easily be adapted, and bigger clusters can be included in the equation.

From equation 8 we also see that the concentration of $H_2SO_4$ is proportional to relative concentrations of sulfuric
acid monomers, dimers and trimers clustered with the bisulphate ion:

$$[H_2SO_4] \sim \frac{[SA_{dimer}] + [SA_{trimer}]}{[SA_{monomer}]} \tag{9}$$


To estimate the $H_2SO_4$ concentration with the ion mode APi-TOF, we can therefore use this theoretical approach,
in particular Eq. 8. For the collision rate of $H_2SO_4$ with $HSO_4^-$ we use $k_1 = 2 \cdot 10^{-9}$ cm$^3$ molecule$^{-1}$ s$^{-1}$ as in Lovejoy
et al. (2004). The value of CS is calculated based on Kulmala et al. (2012). Even if the CS was unknown due, for
example, to the lack of particle measurements, the daytime variability of the $H_2SO_4$ concentration could still be
estimated by using the relation of the $H_2SO_4$-containing cluster with $HSO_4^-$, as it is proportional to the $H_2SO_4$
concentration (see eq. 9). If the concentration of positive small ions is not available, it can be assumed to be in the
range of 500 – 1000 cm$^{-3}$ which is a reasonable approximation for the average concentration (Hirsikko et al.,

170  2011).


As the transmission of clusters within an APi-TOF depends on the tuning of the instrument and on the pressures
within its chambers, the transmission efficiency needs to be considered, in order to get reliable concentrations of
the $SA_{monomer}$, $SA_{dimer}$, and $SA_{trimer}$. Fig. 1 shows the transmission efficiency curve of the APi-TOF used at the
SMEAR II station and Neumayer Station III. The effect of applying the transmission correction to the different

SA clusters is depicted in Fig. 3 for the time series at the SMEAR II station. All ion signals were normalised to a transmission of 1%. As can be determined from Fig. 1a, the $SA_{monomer}$'s transmission at SMEAR II was ~1%, while the dimer and trimer were corrected by a factor of 1/1.8 and 1/1.65, respectively. The correction was also applied on the ions measured at the Neumayer Station III according to the APi-TOF's transmission (Fig. 1b).

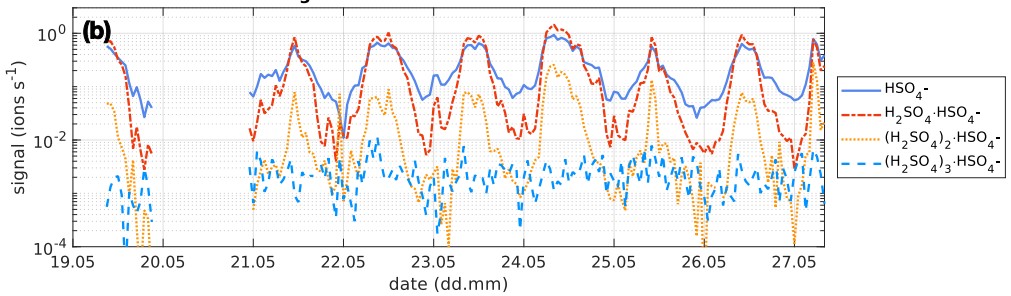

**Figure 3** Time series of the bisulphate ion (HSO$_4^-$, SA$_{monomer}$), H$_2$SO$_4$ clustered with bisulphate (H$_2$SO$_4$·HSO$_4^-$, SA$_{dimer}$), two H$_2$SO$_4$ molecules clustered with the bisulphate ion ((H$_2$SO$_4$)$_2$·HSO$_4^-$, SA$_{trimer}$) and three H$_2$SO$_4$ molecules clustered with the bisulphate ion ((H$_2$SO$_4$)$_3$·HSO$_4^-$, SA$_{tetramer}$) between 19 and 27 May 2017 at SMEAR II station, Hyytiälä, Finland. The concentration is given in ions s$^{-1}$ as measured by the APi-TOF. The upper panel shows the concentration of the clusters considering the transmission efficiency of the instrument (see Fig. 1). The lower panel shows the concentration of the clusters without that correction and assuming a constant transmission efficiency of 1% for all ions.

**3 Validation**

We tested the expression derived above using a dataset collected during inter-comparison measurements at the SMEAR II station in Hyytiälä, Finland (Hari and Kulmala, 2005). In Fig. 4 we show the time series of the observed H$_2$SO$_4$ concentrations, measured with a CI-APi-TOF. The CI-APi-TOF was calibrated for sulfuric acid, based on the method by Kürten et al., (2012) and resulted in a calibration factor of $2.5 \times 10^9$. Additionally, we show the estimated sulfuric acid concentration based on APi-TOF measurements together with Eq. 8 and the sulfuric acid proxy concentration (Dada et al., 2020). The concentration of positive ions for the estimated sulfuric acid concentration was obtained from a Neutral cluster and Air Ion Spectrometer (NAIS, Airel Ltd., Mirme and Mirme, 2013).

The estimated H$_2$SO$_4$ concentration agrees with the measured one during most of the daytime. Between 06:00 and 18:00 local time, the correlation (R$^2$) between the estimated and measured H$_2$SO$_4$ concentration is equal to 0.85

with a root mean square error (RMSE) of $4.12 \times 10^5$ cm$^{-3}$. During night-time, the corresponding values are 0.85
and $3.23 \times 10^5$ cm$^{-3}$ (Table 1).

The scatter plot in Fig. 5 shows that the estimated $H_2SO_4$ concentrations agree well with the observed one when
$H_2SO_4$ concentrations are larger than $2 \times 10^6$ cm$^{-3}$, demonstrating that our method works particularly well at the
SMEAR II station during conditions that favour the formation of $H_2SO_4$-containing clusters.


**Table 1:** Root mean square error (RMSE) and $R^2$ of the estimated $H_2SO_4$ concentration at the SMEAR II station and Neumayer
Station III. The day- and night-time are split in 06:00 – 18:00 local time (LT) and 18:00 – 06:00 LT, respectively. For the
SMEAR II station, we also show the RMSE and $R^2$ of the $H_2SO_4$ proxy calculated with the introduced method by (Dada et al.,

213  2020).

| | Root mean square error (RMSE) | | |
|---|---|---|---|
| | **SMEAR II** | | **Neumayer Station III** |
| | Estimated $H_2SO_4$ eq. (8) | $H_2SO_4$ proxy | Estimated $H_2SO_4$ eq. (8) |
| Daytime | $4.12 \times 10^5$ cm$^{-3}$ | $5.54 \times 10^5$ cm$^{-3}$ | $1.43 \times 10^6$ cm$^{-3}$ |
| Night-time | $3.23 \times 10^5$ cm$^{-3}$ | $4.25 \times 10^5$ cm$^{-3}$ | $1.63 \times 10^6$ cm$^{-3}$ |
| | **$R^2$** | | |
| Daytime | 0.85 | 0.78 | 0.48 |
| Night-time | 0.85 | 0.84 | 0.37 |



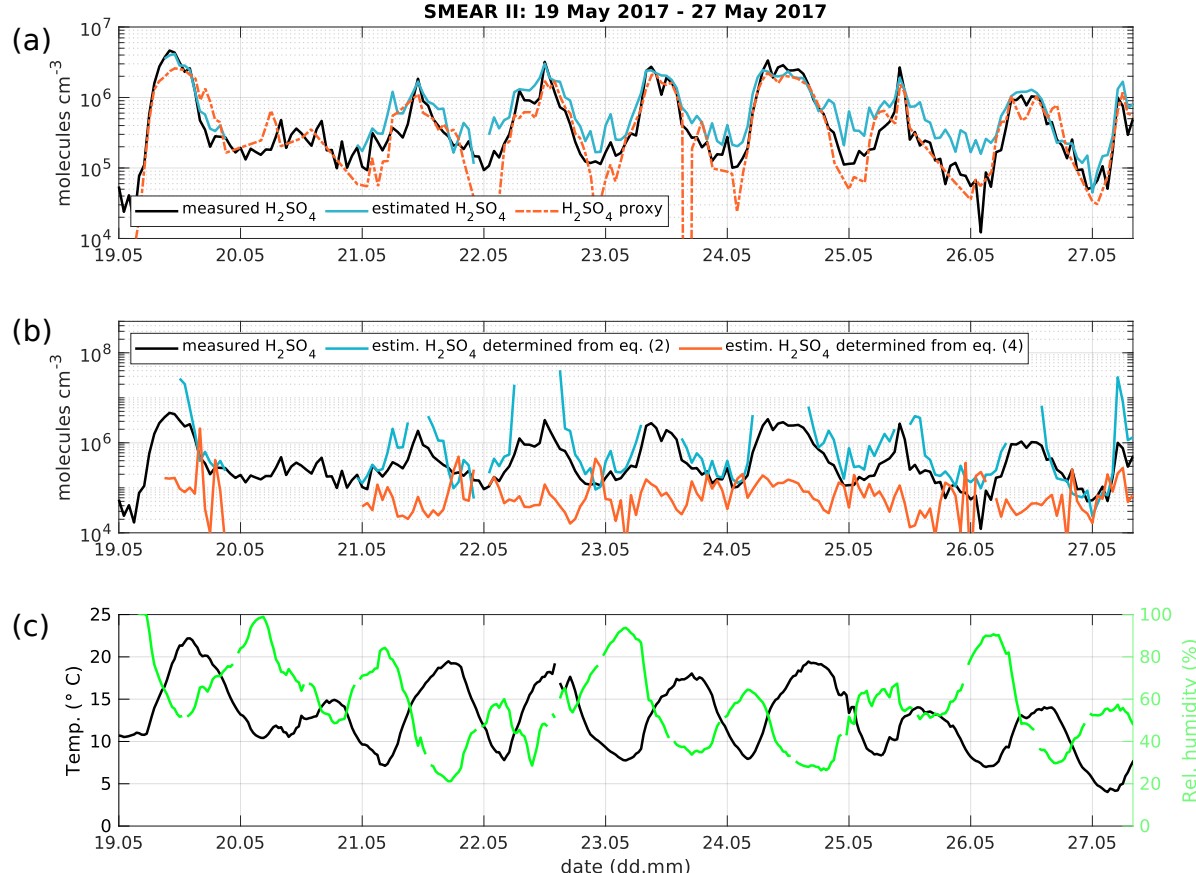


**Figure 4** (a) Time series of measured H$_2$SO$_4$ concentration from the CI-APi-TOF (black) and estimated H$_2$SO$_4$ concentration from the APi-TOF (blue) and H$_2$SO$_4$ proxy from Dada et al. (2020) (orange) between 19 and 27 May 2017. The concentration is given in molecules cm$^{-3}$. (b) Measured H$_2$SO$_4$ concentration as in panel (a) in black and determined concentration from eq. 2 (blue) and eq. 4 (orange). (c) Temperature and relative humidity.

221

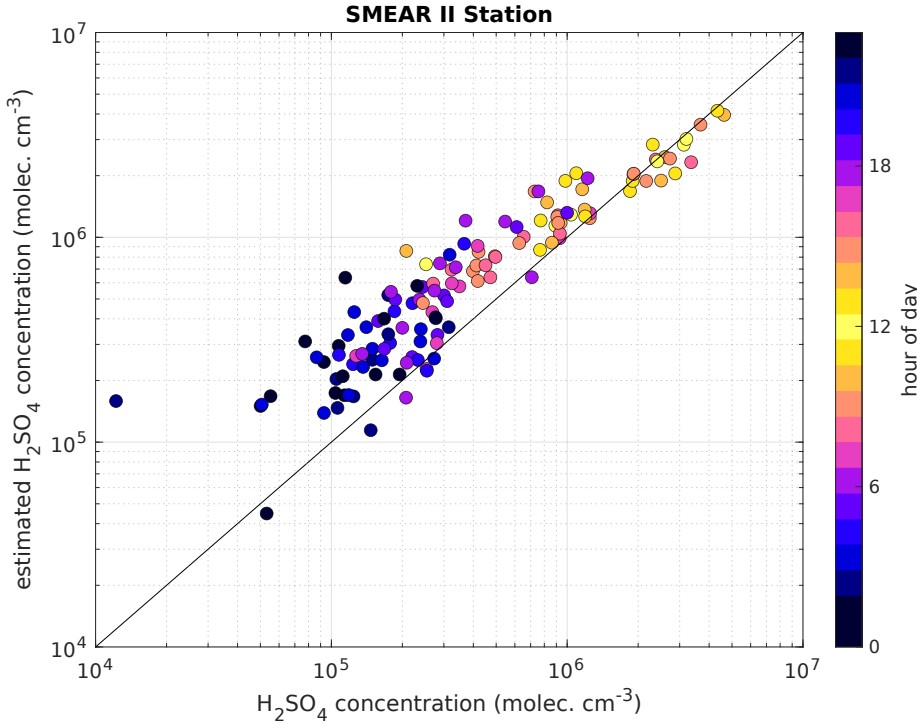

222

**Figure 5** Measured $H_2SO_4$ concentration using a CI-APi-TOF (horizontal-axis) versus estimated $H_2SO_4$ concentration based on APi-TOF results (vertical-axis) at SMEAR II station. For the estimation of $H_2SO_4$, the transmission efficiency was taken into account. The colour is indicating the hour of the day and the black line is the 1:1 ratio. Between 08:00 and 16:00 local time, the concentrations are agreeing well. The shown data contains the time period from 19 to 27 May 2017. The overall correlation coefficient (Pearson) is 0.94.

For the sake of completeness, the estimation of the $H_2SO_4$ concentration determined from Eqs. 2 and 4, assuming pseudo-steady state, are depicted in Fig. 4b. The estimated $H_2SO_4$ concentration from Eq. 2 is overestimating, while solving Eq. 4 for $H_2SO_4$ is underestimating the real concentration as those equations are only approximations. By combining the various approximations, Eq. 8 yields in the best fit to the observed SA concentration.

The presented method was also applied to measurements taken at the Neumayer Station III, Antarctica, in order to test it in a different environment. Here, we used the condensation sink reported by Weller et al. (2015) at Neumayer Station of $1 \times 10^{-3}$ $s^{-1}$. Figure 6 shows a three-week period between 24 December 2018 and 14 January 2019. The calibration factor of the CI-APi-TOF used for measuring the sulfuric acid concentration is $4.9 \times 10^9$. Here, the estimated sulfuric acid concentration underestimates the measured concentration when the $SA_{tetramer}$ and $NH_3(H_2SO_4)_3HSO_4-$ cluster show high concentrations (Fig. 6c). A possible explanation for the underestimation might be the neglection of the growth of sulfuric acid to oligomers larger than the tetramer, as well as its clustering with bases and water (Fig. 6b and c). In coastal Antarctica, the main nucleating mechanism was observed to be negative ion-induced sulfuric acid-ammonia nucleation, acting as a major sink for sulfuric acid molecules due to its clustering with bases (Jokinen et al., 2018). Including the $SA_{tetramer}$ and $SA_{tetramer}$ clustered with $NH_3$ in the estimation equation improved the correlation ($R^2$) from 0.48 to 0.54. Furthermore, as mentioned above, the value of CS for Neumayer was assumed to be constant ($10^{-3}$ $s^{-1}$) due to the lack of data needed for its calculation. This

simplification certainly causes additional errors in estimated SA concentrations, especially during periods of high sea salt concentrations causing potentially large variations in values of CS. Nevertheless, the diurnal variation of the SA concentration is represented well by this method. During times with lower sulfuric acid concentrations, our method gives higher values than the measured concentrations (Fig. 6).

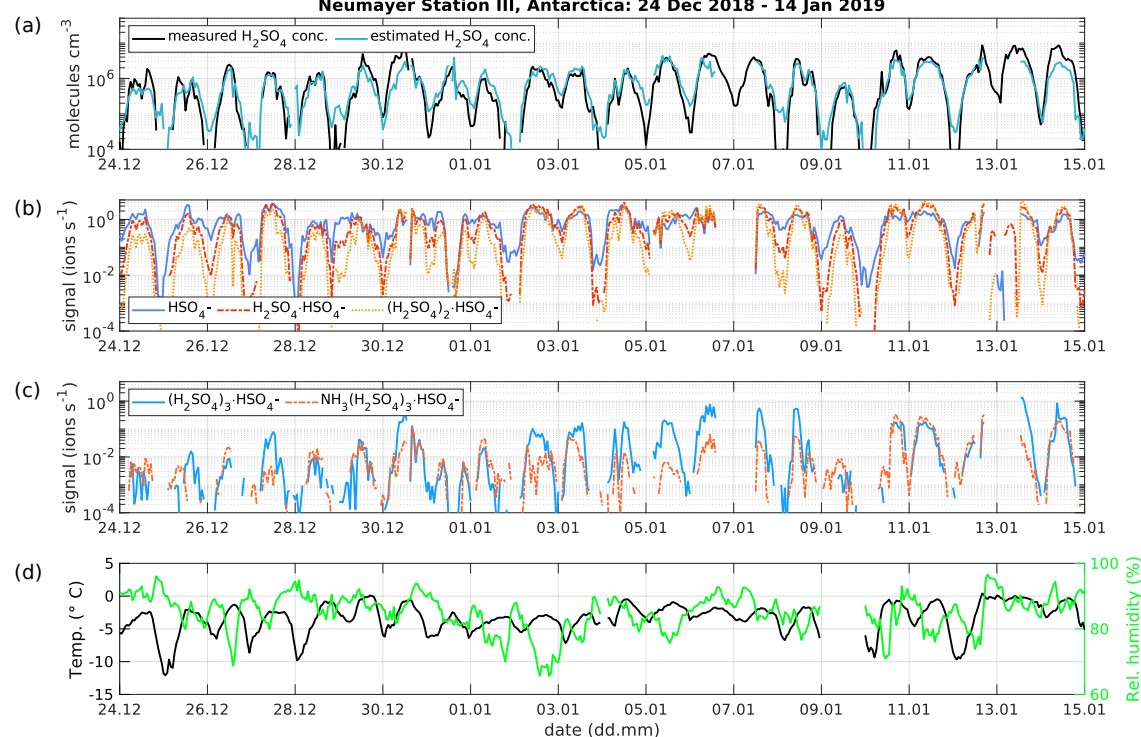

**Figure 6** (a) Time series of measured $H_2SO_4$ concentration from the CI-APi-TOF (black) and estimated $H_2SO_4$ concentration from the APi-TOF (blue) between 24 December 2018 and 14 January 2019 at Neumayer Station III, Antarctica. The concentration is given in molecules $cm^{-3}$. (b) Time series of the bisulphate ion ($HSO_4^-$, $SA_{monomer}$), $H_2SO_4$ clustered with bisulphate ($H_2SO_4 \cdot HSO_4^-$, $SA_{dimer}$), two $H_2SO_4$ molecules clustered with the bisulphate ion (($H_2SO_4)_2 \cdot HSO_4^-$, $SA_{trimer}$) and (c) three $H_2SO_4$ molecules clustered with the bisulphate ion (($H_2SO_4)_3 \cdot HSO_4^-$, $SA_{tetramer}$) as well as the $SA_{tetramer}$ clustered with $NH_3$. (d) Temperature and relative humidity measured at Neumayer Station III.

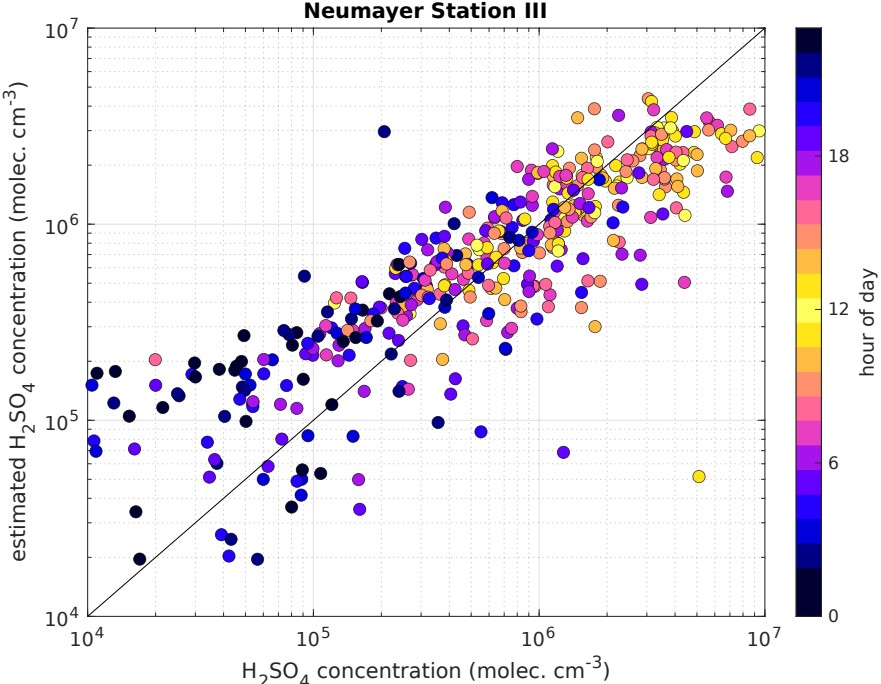

Figure 7 Measured $H_2SO_4$ concentration using a CI-APi-TOF (horizontal axis) versus estimated $H_2SO_4$ concentration based on APi-TOF results (vertical axis) at the Neumayer Station III. For the estimation of $H_2SO_4$, the transmission efficiency was taken into account. The colour is indicating the hour of the day and the black line is the 1:1 ratio. The shown data contains the time period from 24 December 2016 to 14 January 2019. The overall correlation coefficient (Pearson) is 0.77.

## 4 Conclusions

Here we derived a theoretical expression to estimate $H_2SO_4$ concentrations based on APi-TOF measurements of ambient ions. The estimation agrees well with the measured concentration during daytime in the boreal forest ($R^2$ = 0.85), indicating that the estimation is able to represent the diurnal variation and trend of $H_2SO_4$ concentrations during most of the time when active clustering of sulfuric acid is inducing the initial step(s) of atmospheric new particle formation. However, in an atmosphere, where sulfuric acid is the dominating pathway for initiating new particle formation, the method might underestimate the $H_2SO_4$ concentrations, as this method does not include the rapid clustering to bigger of sulfuric acid clusters and clustering with bases directly, e.g. in the Antarctic atmosphere ($R^2$ = 0.48; during daytime).

The APi-TOF's "ion mode", i.e. direct ion sampling without chemical ionisation, remains a crucial tool in many field deployments and laboratory studies, since it is extremely sensitive and allows for observing atmospheric clustering molecule by molecule, which in most cases is impossible when relying on chemical ionization. Therefore, having available a reliable estimate of $H_2SO_4$ concentration allows us to utilise the APi-TOF ion mode even more effectively.

**Data availability**

The data can be accessed via Zenodo (10.5281/zenodo.5266313).

285

**Author contribution**

LJB, SS, VMK and MK designed the study. LJB and MS performed the measurements. SS and LJB derived the equations. LJB processed and analysed the data and performed the data visualisation. MK and VMK supervised the process. All authors commented and edited the paper.

**Competing interests**

The authors declare that they have no conflict of interest.

**Acknowledgements**

We acknowledge the following projects: ACCC Flagship funded by the Academy of Finland grant number 337549, Academy professorship funded by the Academy of Finland (grant no. 302958), Academy of Finland projects no. 1325656, 310682, 316114, 325647 and 296628 , Russian Mega Grant project "Megapolis - heat and pollution island: interdisciplinary hydroclimatic, geochemical and ecological analysis" application reference 2020-220-08-5835, "Quantifying carbon sink, CarbonSink+ and their interaction with air quality" INAR project funded by Jane and Aatos Erkko Foundation, European Research Council (ERC) project ATM-GTP Contract No. 742206 and GASPARCON, grant agreement no. 714621. We thank the tofTools team for providing the tools for the mass spectrometry analysis. We thank the technical and scientific staff in Hyytiälä SMEAR II and the technicians and scientists of the Neumayer overwintering teams of the years 2018 and 2019. We thank Lubna Dada for calculating the SA proxy for SMEAR II station. We thank Janne Lampilahti for providing the codes to process the NAIS dataset.

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
