# Peer review of "Estimation of sulfuric acid concentration using ambient ion"

_Atmospheric Measurement Techniques, 2021_

## Author Comment (AC1)

In this file, the review comments are in black and our responses in green. The added sentences are in *italic*.

RC1

The manuscript "Estimation of sulphuric acid concentrations..." by Lisa J. Beck et al. is generally well written and addresses an important subject in atmospheric research: an approximation method of sulfuric acid concentrations based on sulphuric cluster ion distributions measured by APi-TOF-MS. It is a short paper focussing on deriving one equation and validating it against observations. However, there are some issues with simplifications and the validations should be applied to a wider field of data covering different atmospheric conditions.

We thank the reviewer for the very constructive comments, which we answer below.

General comments:
RC1: "The balance equations (1) to (4) are a simplification probably containing the main processes. However, also with respect to Lovejoy et al. (2004), they do not consider several processes of impact on ambient ions, perhaps most prominent the recombination and the clustering of sulfuric acid ion clusters with water and base molecules. The effect of losses due to recombination with positive ions should be discussed."
AC: This is correct, we do neglect the losses of sulfuric acid to clusters with water and base molecules. We also tested our method with a dataset from Neumayer Station III, Antarctica, where the method is underestimating the actual sulfuric acid concentration, especially on days when a new particle formation event was ongoing. One reason for the underestimation could be the neglect of the clusters of sulfuric acid with bases. We tested the method including also the formation of the tetramer, as it was abundant in a higher concentration than at SMEAR II station. The $R^2$ between 6 – 18 UTC of the estimated sulfuric acid concentration at Neumayer Station III is 0.29 (RMSE: $1.68 \times 10^{-6}$ cm$^{-3)}$) using the presented equation 8 in the manuscript. Including also the tetramer in the estimation equation, the $R^2$ resulted in 0.33 (RMSE: $1.63 \times 10^{-6}$ cm$^{-3}$) and only slightly improved the estimation (note for later discussion: the $R^2$ and RMSE values do **not** include the ion-ion recombination which is discussed below).

For the neglect of the clustering with bases, we added the following information in the manuscript:

*Furthermore, the derivation neglects the losses of SA$_{trimer}$ to the SA$_{tetramer}$ and larger clusters, as well as the clustering of sulfuric acid ion clusters with water and base molecules, such as NH$_3$. Those simplifications can cause an underestimation of the H$_2$SO$_4$ concentration with the presented method. If necessary, the method can easily be adapted, and bigger clusters can be included in the equation.*

As will be discussed below, we included the ion-ion recombination in the revised manuscript. We also included a brief statement regarding the correlation ($R^2$) at Neumayer Station when including the SA$_{tetramer}$ and SA$_{tetramer}$ + NH$_3$ in the method. Since ion-ion recombination is considered in those numbers, they differ from our statement above. We state in the manuscript as follows:

*Including the SAtetramer and SAtetramer clustered with NH₃ in the estimation equation improved the*

*Including the $SA_{tetramer}$ and $SA_{tetramer}$ clustered with $NH_3$ in the estimation equation improved the correlation ($R^2$) from 0.48 to 0.54.*

AC: Further, as the reviewer stated correctly, we did not consider the ion-ion recombination in our presented method which causes additional errors. Therefore, we implemented the losses of charged sulfuric acid clusters due to recombination with positive ions in our equations. We used the equation $\alpha \cdot [SA_i] \cdot N_{pos}]$ (Kontkanen et al., 2013) where alpha is the ion-ion recombination coefficient ($1.6 \times 10^{-6}$ cm$^{-3}$ s$^{-1}$), [SA$_i$] is the concentration of sulfuric acid clusters (monomer, dimer or trimer) and N$_{pos}$ is the concentration of positive small ions. The resulting equation is:

$$[H_2SO_4] = \frac{(CS + \alpha \cdot N_{pos}) \cdot ([SA_{dimer}] + [SA_{trimer}])}{k_1 \cdot [SA_{monomer}]}. \tag{8}$$

We included the ion-ion recombination in our method in the revised manuscript.
For the concentration of N$_{pos}$, we used the measurements with from a Neutral cluster and Air Ion Spectrometer (NAIS, Airel Ltd., Mirme and Mirme, 2013), however if those measurements are not available, N$_{pos}$ can be assumed to be in the range of 500 – 1000 cm$^{-3}$ (Hirsikko et al., 2011).
In the figure below we show the validation of our method on a dataset from Neumayer Station III, Antarctica, and at SMEAR II station with the method presented in our first manuscript (blue solid line) and the method including the ion-ion recombination for the charged sulfuric acid clusters (red dashed line). With consideration of the ion-ion recombination, the estimation of the sulfuric acid concentration has improved at both stations.
At Neumayer Station III, on days when negative ion-induced nucleation was ongoing, e.g. on 14 January 2019, however, the estimated concentration is still underestimating which might be the result of the neglected clustering with bases as discussed above.
The estimation of the sulfuric acid concentration at SMEAR II station also improved during daytime. During night-time, the method is overestimating. The table below shows the RMSE and R$^2$ of the originally presented method and the errors when including the ion-ion recombination for both locations. A table with RMSE and R$^2$ is also included in the revised manuscript.

Another possible reason for the underestimation at Neumayer could be that the CS might be higher than measured due to e.g. intermittent high concentrations of sea salt which cannot be determined reliably. Further, it should be mentioned that the real CS for ions is probably higher than the one used in the usual CS calculations as ions are more likely to condense on pre-existing particles than neutral compounds (Mahfouz and Donahue, 2021). This enhancement of CS has not been taken into account in our calculations, but we state in the manuscript as follows:

"Besides the steady-state assumption, it should be noted that in deriving eq. 8 monomers, dimers and trimers were assumed to have the same loss rate (CS) onto pre-existing aerosol particles. This causes an additional, yet minor, uncertainty in the estimated H$_2$SO$_4$ concentrations, as such loss rates are dependent on the size/mass of the clusters (e.g. Lehtinen et al., 2007; Tuovinen et al.,

2021). *[…] Additionally, the CS for ions is higher than for neutral compounds. The enhancement of CS has shown to reach a maximum value of 2 when the pre-existing particles are < 10 nm and decreases to 1 when the pre-existing particles are > 100 nm, as shown by Mahfouz and Donahue (2021). "*

[Figure]

Time series of measured sulfuric acid concentration (black), the estimated sulfuric acid concentration based on our proposed method (blue) and the estimation including the ion-ion recombination (red) at (a) Neumayer Station III, Antarctica, from 24 December 2018 – 14 January 2019 and (b) SMEAR II, from 19 May – 27 May 2017.

Table 1: Root mean square error and $R^2$ for SMEAR II and Neumayer Station III. The day- and night-time are split in 6 – 18 local time (LT) and 18 – 6 LT, respectively. The root mean square error was calculated for the originally introduced method which neglected the ion recombination and including the recombination. For SMEAR II station, we also show the RMSE and $R^2$ of the $H_2SO_4$ proxy calculated with the introduced method by (Dada et al., 2020).

| | Root mean square error (RMSE) | | | | |
| --- | --- | --- | --- | --- | --- |
| | SMEAR II | | | Neumayer Station III | |
| | Neglecting ion recombination | Including ion recombination | $H_2SO_4$ proxy | Neglecting ion recombination | Including ion recombination |
| Daytime (06–18 LT) | $5.0 \times 10^5$ cm$^{-3}$ | $4.12 \times 10^5$ cm$^{-3}$ | $5.54 \times 10^5$ cm$^{-3}$ | $1.68 \times 10^6$ cm$^{-3}$ | $1.43 \times 10^6$ cm$^{-3}$ |
| Night-time (18–06 LT) | $3.54 \times 10^5$ cm$^{-3}$ | $3.23 \times 10^5$ cm$^{-3}$ | $4.25 \times 10^5$ cm$^{-3}$ | $2.54 \times 10^7$ cm$^{-3}$ | $1.63 \times 10^6$ cm$^{-3}$ |
| | $R^2$ | | | | |
| Daytime (06–18 LT) | 0.78 | 0.85 | 0.78 | 0.29 | 0.48 |
| Night-time (18–06 LT) | 0.83 | 0.85 | 0.84 | -154 | 0.37 |

[Figure]

Measured (horizontal axis) and estimated (vertical axis) sulfuric acid concentration from Neumayer Station III, Antarctica. The left figure (a) shows the method without ion-ion recombination, the figure on the right (b) shows the method including the ion-ion recombination. The colouring indicates the hour of the day (local time).

[Figure]

Measured (horizontal axis) and estimated (vertical axis) sulfuric acid concentration from SMEARII, Finland. The figure on the left (a) shows the method without ion-ion recombination, the figure on the right (b) shows the method including the ion-ion recombination. The colouring indicates the hour of the day (local time).

RC1: "Further, the APi-TOF may not show real ambient ion clusters as in the process of pumping away neutral molecules and transfer of ions into the high vacuum TOF region, weakly bound molecules are expected to be dissociated from the clusters in collisions. And condensation sink is, as correctly stated, expected to be dependent on mass and size of the clusters. Yet, effects are expected to be minor but should be discussed."

AC: The APi-TOF only detects roughly 1% or less of the actual ambient ion cluster concentration (Junninen et al., 2010). In order to quantify how many ions are reaching the detector of the APi-TOF, we included the transmission efficiency calibration curve in the manuscript. Here, the concentration of ions measured by APi-TOF is compared to the concentration measured with an electrometer for different size ranges. We included the correction of the different ions (SA$_{monomer}$, SA$_{dimer}$, and SA$_{trimer}$) accordingly for the validation of the method, which improved the outcome of the estimated SA concentration.

[Figure]

**Figure 1** Ion transmission of the APi-TOFs used in this study. The transmission efficiency was determined via production of charged particles with a NiCr wire. The concentration of the size selected ions with a Hermann nano differential mobility analyser (HDMA, Hermann, 2000) were measured with an electrometer and an APi-TOF in parallel. A more detailed description can be found in Junninen et al. (2010). Panel (a) shows the transmission efficiency of the APi-TOF used for measurements at SMEAR II Station, Hyytiälä, Finland. Panel (b) shows the transmission efficiency used for measurements at Neumayer Station III.

In our study, we did not quantify the fragmentation of weakly bound molecules within the APi-TOF. The fragmentation due to the voltage of the electrodes within the APi-TOF have been studied by Passananti et al. (2019). The charged sulfuric acid dimer is a very stable cluster with an evaporation rate of $2.7 \times 10^{-15}$ s$^{-1}$, while the charged sulfuric acid trimer is a little less stable with an evaporation rate of $5.6 \times 10^{-3}$ s$^{-1}$ (Ortega et al., 2014). If a charged sulfuric acid trimer is fragmented, it loses a sulfuric acid molecule and will be detected as a sulfuric acid dimer. In our method, this would not affect the estimated concentration, as the numerator contains the sum of sulfuric acid dimer and trimer. However, the fragmentation of bigger clusters and clusters with bases might affect the estimated sulfuric acid concentration.

The CS is dependent on the composition, mass and size of the cluster. We included a short paragraph in the manuscript as follows:

*According to Tuovinen et al. (2021), the CS of $H_2SO_4$ clusters decreases with increasing number of $H_2SO_4$ molecules. The study shows that the CS of the $SA_{dimer}$ clustered with ammonia decreases to 68% (compared to one $H_2SO_4$ molecule) and for $SA_{pentamer}$ with four ammonia molecules to 42%. However, the order of magnitude of the CS remains the same, and the effect on the estimation of the $H_2SO_4$ concentration is assumed to be negligible.*

RC1:

The made simplifications give rise to the following issue: each budget equation, excluding eq.(1), can be solved for H2SO4 on itself. In pseudo-steady state, (2) then yields

[H2SO4] = CS [SAdimer] / (k1 [SAmonomer] - k2 [SAdimer]) And (3) yields:

[H2SO4] = CS [SAtrimer] / k2 [SAdimer]

The constant k2 can be estimated from Lovejoy et al. (2004) to be very close to k1.

Thus, together with eq. (8) of the manuscript, three equations to determine H2SO4 can be derived. Obviously, these yield different approximations of H2SO4. The differences are due to incomplete balances and the made assumptions. It is recommended and expected that the authors discuss the corresponding differences.

AC: The calculated sulfuric acid concentration from equation 2 and equation 4 were added to the manuscript and briefly discussed. As expected, the values resulting from equation 2 are highly overestimating, while the results from equation 4 underestimating the sulfuric acid concentration in both locations where we validated the method. We state in the manuscript:

*For the sake of completeness, the estimation of the $H_2SO_4$ concentration determined from Eqs. 2 and 4, assuming pseudo-steady state, are depicted in Fig. 4b. The estimated $H_2SO_4$ concentration from Eq. 2 is highly overestimating, since the losses of the $SA_{dimer}$ to the $SA_{trimer}$ are neglected. When solving Eq. 4 for $H_2SO_4$, only the needed $H_2SO_4$ for the formation of the trimer is considered and the monomer and dimer production are neglected. Consequently, the resulting estimated $H_2SO_4$ concentration is vastly underestimating the real concentration.*

We added the results of the estimated sulfuric acid concentration in the former Fig. 2 (now Fig. 4):

[Figure]

**Figure 4** (a) Time series of measured H$_2$SO$_4$ concentration from the CI-APi-TOF (black) and estimated H$_2$SO$_4$ concentration from the APi-TOF (blue), estimated H$_2$SO$_4$ concentration including ion-ion recombination (red) and H$_2$SO$_4$ proxy from Dada et al. (2020) (orange dashed) between 19 and 28 May 2017. The concentration is given in molecules cm$^{-3}$. (b) Measured H$_2$SO$_4$ concentration as in panel (a) in black and determined concentration from eq. 2 (blue) and eq. 4 (orange). (c) Temperature and relative humidity.

RC1: In section 3 "Validation" the estimated and measured concentrations of a period of 8 days are compared. Though I agree that above 2x10$^6$ molecules cm$^{-3}$ agreement is good in this logarithmic presentation, there is also a period starting in the evening of May 26 with larger deviations. Together with some night-time overestimations of the approximation, there remains the question if the agreement in the five consecutive days 19-25 May was achieved accidentally. It is recommended to discuss this question. From Fig. 3, the trimer makes the difference in the last period, is there any explanation? Overall, recommending the applicability of eq. (8) for general use appears premature and will need further proof that eq. (8) can yield reasonable estimates under varying temperature, humidity and pressure conditions.

AC: In order to test the method in a different environment, we applied it on a three-week dataset from Neumayer Station III in Antarctica and added the results in the manuscript. The estimated sulfuric acid concentration is representing the measured sulfuric acid concentration quite well and captures the diurnal variation. However, the method is underestimating the concentration on some

days. One reason therefore is most likely due to the neglect of the formation of bigger oligomers than the trimer as well as the clustering with other bases. Therefore, we show the time series of the sulfuric acid monomer, dimer, trimer, tetramer and sulfuric acid tetramer clustered with $NH_3$. From the time series we can determine that the method is specifically underestimating on days when the $SA_{tetramer}$ and $NH_3(H_2SO_4)_3HSO_4-$ concentrations are high. We tested if the method can be improved by including more oligomers (tetramer and pentamer of sulfuric acid), however it did not improve the estimation of sulfuric acid significantly: the correlation ($R^2$) changed from 0.48 (without $SA_{tetramer}$ and $SA_{tetramer}$-$NH_3$ cluster) to 0.54 (including $SA_{tetramer}$ and $SA_{tetramer}$-$NH_3$ cluster). Still, our proposed method can easily be adapted, and bigger clusters can be included in the estimation method if needed.

As discussed previously, the neglect of ion-ion recombination in our originally proposed method causes additional errors in the estimation of the concentration. As stated above, including the recombination of the negatively charged SA cluster considerably improved the method. We therefore include the ion-ion recombination in the revised manuscript and in the presented figure below.

[Figure]

**Figure 6** (a) Time series of measured $H_2SO_4$ concentration from the CI-APi-TOF (black), estimated $H_2SO_4$ concentration from the APi-TOF (blue), and estimated $H_2SO_4$ concentration including ion-ion recombination (red) between 24 December 2018 and 15 January 2019 at Neumayer Station III, Antarctica. The concentration is given in molecules cm$^{-3}$. (b) Time series of the bisulphate ion (HSO$_4^-$, $SA_{monomer}$), $H_2SO_4$ clustered with bisulphate ($H_2SO_4 \cdot HSO_4-$, $SA_{dimer}$), two $H_2SO_4$ molecules clustered with the bisulphate ion (($H_2SO_4)_2 \cdot HSO_4-$, $SA_{trimer}$) and (c) three $H_2SO_4$ molecules clustered with the bisulphate ion (($H_2SO_4)_3 \cdot HSO_4-$, $SA_{tetramer}$) as well as the $SA_{tetramer}$ clustered with $NH_3$. (d) Temperature and relative humidity measured at Neumayer Station III.

Specific comments:

RC1: L. 20-23: It is recommended to be more careful in claiming the theoretical expression for H2SO4 may be used under various atmospheric conditions (see also general comment 3)

AC: we rephrased in the Abstract to:

*Here, we propose a theoretical method to estimate the SA concentration based on ambient ion composition and concentration measurements that are achieved by APi-TOF alone.*

RC1: L. 24: "developed estimate works very well..." is a rather qualitative description, better quantify by objective measures.

AC: We included the root mean square error and correlation ($R^2$) in the manuscript, to be more precise. The values are shown as table and also included in the revised text. The finalised table of the revised manuscript is shown below:

**Table 1:** Root mean square error (RMSE) and $R^2$ of the estimated $H_2SO_4$ concentration at the SMEAR II station and Neumayer Station III. The day- and night-time are split in 06:00 – 18:00 local time (LT) and 18:00 – 06:00 LT, respectively. For the SMEAR II station, we also show the RMSE and $R^2$ of the $H_2SO_4$ proxy calculated with the introduced method by (Dada et al., 2020).

| | Root mean square error (RMSE) | | |
| --- | --- | --- | --- |
| | SMEAR II | | Neumayer Station III |
| | Estimated $H_2SO_4$ eq. (8) | $H_2SO_4$ proxy | Estimated $H_2SO_4$ eq. (8) |
| Daytime | $4.12 \times 10^5$ cm$^{-3}$ | $5.54 \times 10^5$ cm$^{-3}$ | $1.43 \times 10^6$ cm$^{-3}$ |
| Night-time | $3.23 \times 10^5$ cm$^{-3}$ | $4.25 \times 10^5$ cm$^{-3}$ | $1.63 \times 10^6$ cm$^{-3}$ |
| | $R^2$ | | |
| Daytime | 0.85 | 0.78 | 0.48 |
| Night-time | 0.85 | 0.84 | 0.37 |

RC1: L 29-36: Some credit should be given to early ambient ion distribution and sulphuric acid measurements by the Eisele and Arnold groups.

AC: We included Arnold and Fabian (1980) and Eisele (1989) in the introduction as follows:

*A first attempt of estimating the sulfuric acid concentration via the concentration of atmospheric ions was introduced by Arnold and Fabian (1980), followed by Eisele (1989) under the assumption that most $H_2SO_4$ molecules are charged by reacting with $NO_3^-$.*

RC1: L. 44-48: It is unclear if this is just an estimate or based on experimentally determined detection limits of the described system. Please, be clearer.

AC: We included the transmission efficiency calibration results of the two APi-TOFs used in the study (one at SMEAR II station, one at Neumayer Station, which was now added in the revision process). The transmission curves show, that 1-1.8% of the ions with a mass smaller than 400 Th are transmitted through the APi-TOF used at SMEAR II station and $0.5 - 1\%$ of the ions with a mass of $100 - 500$ Th are transmitted through the APi-TOF used at Neumayer Station III.

[Figure]

RC1: L. 60-62: Proxies for H2SO4 and the here presented estimation based on atmospheric ions are both using several assumptions and it is not clear which approach is better under which conditions. Thus, either demonstrate results by both approaches and judge the agreement or be more cautious in presenting an advantage of the new approach, which is rather tentative.

AC: We rephrase the statement to be more cautious to:

*In circumstances, where the required data for $H_2SO_4$ proxies is not available, but measurements with an APi-TOF were conducted, the $H_2SO_4$ concentration can be obtained by the ion mass spectra.*

RC1: L. 74: "... we theoretically explain ...". Here, and later on, I'd recommend to be more careful in the wording, as the derived formula is an approximation of the H2SO4 concentrations based on the ion abundances.

AC: The sentence was corrected to:

*In order to estimate the sulfuric acid concentration ($H_2SO_4$) using measured naturally charged ions (see Fig. 2), we approximate this concentration by following the bisulphate ion $HSO_4^-$, herein denoted $SA_{monomer}$, the dimer cluster $H_2SO_4 \cdot HSO_4^-$ ($SA_{dimer}$) and trimer cluster $(H_2SO_4)_2 \cdot HSO_4^-$ ($SA_{trimer}$).*

RC1: Conclusion: It is recommended to be more cautious and precise and avoid "...give accurate enough..." and "... a reliable estimate...".

AC: We rephrased the text in order to be more precise and considering the added dataset from Antarctic atmosphere. We also included the values of correlation in the conclusions:

*The estimation agrees well with the measured concentration during daytime in the boreal forest ($R^2 = 0.85$), indicating that the estimation is able to represent the diurnal variation and trend of $H_2SO_4$ concentrations during most of the time when active clustering of sulfuric acid is inducing the initial step(s) of atmospheric new particle formation. However, in an atmosphere, where sulfuric acid is the dominating pathway for initiating new particle formation, the method might underestimate the $H_2SO_4$ concentrations, as this method does not include the rapid clustering to bigger of sulfuric acid clusters and clustering with bases directly, e.g. in the Antarctic atmosphere ($R^2 = 0.48$; during daytime).*

References:

Dada, L., Ylivinkka, I., Baalbaki, R., Li, C., Guo, Y., Yan, C., Yao, L., Sarnela, N., Jokinen, T., Daellenbach, K. R., Yin, R., Deng, C., Chu, B., Nieminen, T., Wang, Y., Lin, Z., Thakur, R. C., Kontkanen, J., Stolzenburg, D., Sipilä, M., Hussein, T., Paasonen, P., Bianchi, F., Salma, I., Weidinger, T., Pikridas, M., Sciare, J., Jiang, J., Liu, Y., Petäjä, T., Kerminen, V.-M., and Kulmala, M.: Sources and sinks driving sulfuric acid concentrations in contrasting environments: implications on proxy calculations, 20, 11747–11766, https://doi.org/10.5194/acp-20-11747-2020, 2020.

Herrmann, W., Eichler, T., Bernardo, N., and Fernandez de la Mora, J.: Turbulent transition arises at Re 35 000 in a short Vi- enna type DMA with a large laminarizing inlet, Proceedings of the annual conference of the AAAR, St. Louis, MO, 6–10 Octo- ber 2000.

Hirsikko, A., Nieminen, T., Gagné, S., Lehtipalo, K., Manninen, H. E., Ehn, M., Hõrrak, U., Kerminen, V.-M., Laakso, L., McMurry, P. H., Mirme, A., Mirme, S., Petäjä, T., Tammet, H., Vakkari, V., Vana, M., and Kulmala, M.: Atmospheric ions and nucleation: a review of observations, 11, 767–798, https://doi.org/10.5194/acp-11-767-2011, 2011.

Kontkanen, J., Lehtinen, K. E. J., Nieminen, T., Manninen, H. E., Lehtipalo, K., Kerminen, V.-M., and Kulmala, M.: Estimating the contribution of ion–ion recombination to sub-2 nm cluster concentrations from atmospheric measurements, 13, 11391–11401, https://doi.org/10.5194/acp-13-11391-2013, 2013.

Kürten, A., Rondo, L., Ehrhart, S., and Curtius, J.: Calibration of a Chemical Ionization Mass Spectrometer for the Measurement of Gaseous Sulfuric Acid, J. Phys. Chem. A, 116, 6375–6386, https://doi.org/10.1021/jp212123n, 2012.

Lehtinen, K. E. J., Dal Maso, M., Kulmala, M., and Kerminen, V.-M.: Estimating nucleation rates from apparent particle formation rates and vice versa: Revised formulation of the Kerminen–Kulmala equation, Journal of Aerosol Science, 38, 988–994, https://doi.org/10.1016/j.jaerosci.2007.06.009, 2007.

Mahfouz, N. G. A. and Donahue, N. M.: Technical note: The enhancement limit of coagulation scavenging of small charged particles, 21, 3827–3832, https://doi.org/10.5194/acp-21-3827-2021, 2021.

Mirme, S. and Mirme, A.: The mathematical principles and design of the NAIS – a spectrometer for the measurement of cluster ion and nanometer aerosol size distributions, 6, 1061–1071, https://doi.org/10.5194/amt-6-1061-2013, 2013.

Tuovinen, S., Kontkanen, J., Cai, R., and Kulmala, M.: Condensation sink of atmospheric vapors: the effect of vapor properties and the resulting uncertainties, Environ. Sci.: Atmos., 1, 543–557, https://doi.org/10.1039/D1EA00032B, 2021.

---

## Author Comment (AC2)

In this file, the review comments are in black and our responses in green. The added sentences are in *italic*.

**General comments:**

RC2: The manuscript by Beck at al. presents the derivation of an equation to approximate sulfuric acid concentrations in the atmosphere using APi-TOF data when CI is not available. The manuscript is fairly short and as such focuses on a narrow topic, albeit an important parameter that can be observed in the atmosphere by (CI-)APi-TOF. I understand that this can be of use to the community of APi-TOF users and for measurements of sulfuric acid in new particle formation studies, and would therefore support publication after reviewer comments have been addressed. I would however have hoped for a somewhat more comprehensive study and especially evaluation of the proposed approximation, and would encourage the authors to expand especially the validation section with more datasets, which must be available to them. How well does their approximation compare to the other proxies mentioned in the introduction? Are there other data than CI-APi-TOF data available to validate the approximation?

We thank the reviewer for the very constructive comments, which we answer below.

AC: We included the sulfuric acid proxy from Dada et al. (2020) for the SMEAR II dataset.

Further, we included a three-week period from Neumayer Station III to validate the method. Unfortunately, we did not have enough variables to calculate the sulfuric proxy for Neumayer Station III.

The validation shows, that at Neumayer Station, the estimation with our method is underestimating the concentration during daytime. But, as reviewer 1 stated correctly, we neglected the ion-ion recombination in our method. By including the recombination term, the correlation ($R^2$) of the estimated sulfuric acid concentration during daytime improved from 0.29 to 0.48. The term in the equation for ion-ion recombination is especially relevant in atmospheres, where the CS is low, like for example at Neumayer Station III.

On days, when the concentration of larger sulfuric acid clusters and clusters of sulfuric acid and a base is high and nucleation was ongoing, the estimated concentration was however still too low (e.g. 14 January 2019 at Neumayer Station). Therefore, we conclude, that the neglection of those clusters causes errors, especially in atmospheres, where the main mechanism for new particle formation is involving negative ion induced sulfuric acid nucleation (Jokinen et al., 2018).

In the table below, we show the correlation and root mean square errors of our previously presented method (neglecting ion recombination) as well as the renewed method including ion-ion recombination and the SA proxy (only for SMEAR II station).

Table 1: Root mean square error (RMSE) and $R^2$ for SMEAR II and Neumayer Station III. The day- and night-time are split in 6 – 18 local time (LT) and 18 – 6 LT, respectively. The root mean square error was calculated for the originally introduced method which neglected the ion recombination and including the recombination. For SMEAR II station, we also show the RMSE and $R^2$ of the $H_2SO_4$ proxy calculated with the introduced method by (Dada et al., 2020).

| | Root mean square error (RMSE) | | | | |
| --- | --- | --- | --- | --- | --- |
| | SMEAR II | | | Neumayer Station III | |
| | Neglecting ion recombination | Including ion recombination | $H_2SO_4$ proxy | Neglecting ion recombination | Including ion recombination |
| Daytime (06–18 LT) | $5.0 \times 10^5$ cm$^{-3}$ | $4.12 \times 10^5$ cm$^{-3}$ | $5.54 \times 10^5$ cm$^{-3}$ | $1.68 \times 10^6$ cm$^{-3}$ | $1.43 \times 10^6$ cm$^{-3}$ |
| Night-time (18–06 LT) | $3.54 \times 10^5$ cm$^{-3}$ | $3.23 \times 10^5$ cm$^{-3}$ | $4.25 \times 10^5$ cm$^{-3}$ | $2.54 \times 10^7$ cm$^{-3}$ | $1.63 \times 10^6$ cm$^{-3}$ |
| | $R^2$ | | | | |
| Daytime (06–18 LT) | 0.78 | 0.85 | 0.78 | 0.29 | 0.48 |
| Night-time (18–06 LT) | 0.83 | 0.85 | 0.84 | -154 | 0.37 |

The figure below shows the validation of the estimated SA concentration for Neumayer Station III. The time series in panel (a) shows the resulting SA concentration with our previously suggested method (blue solid line) and including ion-ion recombination (red dashed line). For the revised manuscript, we included the ion-ion recombination, as suggested by reviewer 1.

[Figure]

**Figure 6** (a) Time series of measured $H_2SO_4$ concentration from the CI-APi-TOF (black), estimated $H_2SO_4$ concentration from the APi-TOF (blue), and estimated $H_2SO_4$ concentration including ion-ion recombination (red) between 24 December 2018 and 15 January 2019 at Neumayer Station III, Antarctica. The concentration is given in molecules cm$^{-3}$. (b) Time series of the bisulphate ion ($HSO_4^-$, $SA_{monomer}$), $H_2SO_4$ clustered with bisulphate ($H_2SO_4 \cdot HSO_4^-$, $SA_{dimer}$), two $H_2SO_4$ molecules clustered with the bisulphate ion (($H_2SO_4)_2 \cdot HSO_4^-$, $SA_{trimer}$) and (c) three $H_2SO_4$ molecules clustered with the bisulphate ion (($H_2SO_4)_3 \cdot HSO_4^-$

Unfortunately, we do not have other data available than from CI-APi-TOF to compare our method to.

RC2: Figure 1 presents a spectrum in ions per second (the captions mentions time and day but not location, this should be added),

AC: we added the location, and also a spectrum from Antarctica, as data from there was included for further validation.

RC2: Figures 2 and 4 molecules per cm3. No information is given on conversion factors or sensitivity assumptions. This is especially important when comparing CI-APi-TOF and APi-TOF. Related to that, how reproducible are ratios of SA monomers, dimers, trimers between APi-TOF instruments? Can it be assumed that all clusters are detected with equal sensitivity? Can the authors elaborate on that?

AC: We added the transmission calibration of the used APi-TOFs as a figure and implemented the transmission efficiency of each ion in the calculations. We state in the manuscript:

*As the transmission of clusters within an APi-TOF depends on the tuning of the instrument and on the pressures within its chambers, the transmission efficiency needs to be considered, in order to get reliable concentrations of the $SA_{monomer}$, $SA_{dimer}$, and $SA_{trimer}$. Fig. 1 shows the transmission efficiency curve of the APi-TOF used at the SMEAR II station and Neumayer Station III. The effect of applying the transmission correction to the different SA clusters is depicted in Fig. 3 for the time series at the SMEAR II station. All ion signals were normalised to a transmission of 1%. As can be determined from Fig. 1a, the $SA_{monomer}$'s transmission at SMEAR II was ~1%, while the dimer and trimer were corrected by a factor of 1/1.8 and 1/1.65, respectively. The correction was also applied on the ions measured at the Neumayer Station III according to the APi-TOF's transmission (Fig. 1b).*

[Figure]

**Figure 1** Ion transmission of the APi-TOFs used in this study. The transmission efficiency was determined via production of charged particles with a NiCr wire. The concentration of the size selected ions with a Hermann nano differential mobility analyser (HDMA, Hermann, 2000) were measured with an electrometer and an APi-TOF in parallel. A more detailed description can be found in Junninen et al. (2010). Panel (a) shows the transmission efficiency of the APi-TOF used for measurements at the SMEAR II Station, Hyytiälä, Finland. Panel (b) shows the transmission efficiency used for measurements at the Neumayer Station III.

[Figure]

**Figure 3** Time series of the bisulphate ion (HSO$_4^-$, SA$_{monomer}$), H$_2$SO$_4$ clustered with bisulphate (H$_2$SO$_4$·HSO$_4$-, SA$_{dimer}$), two H$_2$SO$_4$ molecules clustered with the bisulphate ion ((H$_2$SO$_4$)$_2$·HSO$_4$-, SA$_{trimer}$) and three H$_2$SO$_4$ molecules clustered with the bisulphate ion ((H$_2$SO$_4$)$_3$·HSO$_4$-, SA$_{tetramer}$) between 19 and 27 May 2017 at SMEAR II station, Hyytiälä, Finland. The concentration is given in ions s$^{-1}$ as measured by the APi-TOF. The upper panel shows the concentration of the clusters considering the transmission efficiency of the instrument (see Fig. 1). The lower panel shows the concentration of the clusters without that correction and assuming a constant transmission efficiency of 1% for all ions.

Further, we added the calibration factor for both CI-APi-TOFs in the text. The CI-APi-TOFs were calibrated with sulfuric acid, based on the proposed method by Kürten et al. (2012). The calibration factor of the CI-APi-TOF at SMEAR II station is 2.5x10$^9$ and 4.9x10$^9$ for the CI-APi-TOF at Neumayer Station III.

**Technical comments:**

Title: Since the technique is mentioned as abbreviation in the title, I suggest using its full name in the title as well.

We rephrased the title as suggested to: *"Estimation of sulfuric acid concentrations using ambient ion composition and concentration data obtained by Atmospheric Pressure interface Time-of-Flight ion mass spectrometer"*

RC2: Line 18 – 19: Move "CI" behind "chemical ionization" and remove it from "(CI-APi-TOF)"

AC: We added the CI abbreviation behind "chemical ionization", as requested.

RC2: Line 33: "detect this concentration" – at all? Quantitatively? Can the authors be more specific?

We added the following text (written italic) to be more specific about the challenges regarding the measurement of sulfuric acid:

However, ambient concentrations of $H_2SO_4$ are low, commonly less than a part per trillion by volume (~$2 \cdot 10^7$ molecules cm$^{-3}$), making it challenging to *measure it*. During the recent years there have been instrumental developments towards a reliable detection of $H_2SO_4$ in the atmosphere, particularly via the development of a Chemical Ionisation Atmospheric Pressure interface Time-of-Flight mass spectrometer (CI-APi-TOF, Jokinen et al., 2012), using nitric acid as a reagent ion. *Still, the measurement technique with CI-APi-TOF is relatively challenging, as a thorough calibration i.e. with sulfuric acid as proposed by Kürten et al. (2012), is needed in order to get reliable numbers. Furthermore, the loss of sulfuric acid to surfaces, such as an inlet, and the correct flow rates must be known and characterised.*

RC2: Line 34: Not clear to me what is meant by "clear steps"

AC: we rephrased the sentence to:

During the recent years there have been *instrumental developments* towards a reliable detection of $H_2SO_4$ in the atmosphere, particularly via the development of a Chemical Ionisation Atmospheric Pressure interface Time-of-Flight mass spectrometer (CI-APi-TOF, Jokinen et al., 2012), using nitric acid as a reagent ion.

RC2: Line 38: "During the past decade or so" sounds rather colloquial, suggest rewording

AC: We removed the phrase "or so" and introduce the sentence with:

*During the past decade, ...*

RC2: Line 46: should read "ions"

AC: Thank you for noticing the correction is implemented in the reviewed manuscript.

RC2: Figure 1: Red peaks could be labelled individually

AC: We added labels to the ions in the figure with the spectra (see figure below).

[Figure]

Figure 2 (a) Mass spectrum from 50 to 600 Th measured with the APi-TOF on 24 May 2017 during the time period 08:00 – 18:00 (local time) at SMEAR II station, Hyytiälä, Finland. (b) Mass spectrum from 14 January 2019 between 08:00 and 18:00 (local time) at Neumayer Station III, Antarctica during a new particle formation event. The bisulphate ion HSO4- and H2SO4 clusters containing it were used for the estimation of $H_2SO_4$ concentration, and are coloured in red.

References:

Dada, L., Ylivinkka, I., Baalbaki, R., Li, C., Guo, Y., Yan, C., Yao, L., Sarnela, N., Jokinen, T., Daellenbach, K. R., Yin, R., Deng, C., Chu, B., Nieminen, T., Wang, Y., Lin, Z., Thakur, R. C., Kontkanen, J., Stolzenburg, D., Sipilä, M., Hussein, T., Paasonen, P., Bianchi, F., Salma, I., Weidinger, T., Pikridas, M., Sciare, J., Jiang, J., Liu, Y., Petäjä, T., Kerminen, V.-M., and Kulmala, M.: Sources and sinks driving sulfuric acid concentrations in contrasting environments: implications on proxy calculations, 20, 11747–11766, https://doi.org/10.5194/acp-20-11747-2020, 2020.

Herrmann, W., Eichler, T., Bernardo, N., and Fernandez de la Mora, J.: Turbulent transition arises at Re 35 000 in a short Vi- enna type DMA with a large laminarizing inlet, Proceedings of the annual conference of the AAAR, St. Louis, MO, 6–10 Octo- ber 2000.

Hirsikko, A., Nieminen, T., Gagné, S., Lehtipalo, K., Manninen, H. E., Ehn, M., Hõrrak, U., Kerminen, V.-M., Laakso, L., McMurry, P. H., Mirme, A., Mirme, S., Petäjä, T., Tammet, H., Vakkari, V., Vana, M., and Kulmala, M.: Atmospheric ions and nucleation: a review of observations, 11, 767–798, https://doi.org/10.5194/acp-11-767-2011, 2011.

Kontkanen, J., Lehtinen, K. E. J., Nieminen, T., Manninen, H. E., Lehtipalo, K., Kerminen, V.-M., and Kulmala, M.: Estimating the contribution of ion–ion recombination to sub-2 nm cluster concentrations from atmospheric measurements, 13, 11391–11401, https://doi.org/10.5194/acp-13-11391-2013, 2013.

Kürten, A., Rondo, L., Ehrhart, S., and Curtius, J.: Calibration of a Chemical Ionization Mass Spectrometer for the Measurement of Gaseous Sulfuric Acid, J. Phys. Chem. A, 116, 6375–6386, https://doi.org/10.1021/jp212123n, 2012.

Lehtinen, K. E. J., Dal Maso, M., Kulmala, M., and Kerminen, V.-M.: Estimating nucleation rates from apparent particle formation rates and vice versa: Revised formulation of the Kerminen–Kulmala equation, Journal of Aerosol Science, 38, 988–994, https://doi.org/10.1016/j.jaerosci.2007.06.009, 2007.

Mahfouz, N. G. A. and Donahue, N. M.: Technical note: The enhancement limit of coagulation scavenging of small charged particles, 21, 3827–3832, https://doi.org/10.5194/acp-21-3827-2021, 2021.

Mirme, S. and Mirme, A.: The mathematical principles and design of the NAIS – a spectrometer for the measurement of cluster ion and nanometer aerosol size distributions, 6, 1061–1071, https://doi.org/10.5194/amt-6-1061-2013, 2013.

Tuovinen, S., Kontkanen, J., Cai, R., and Kulmala, M.: Condensation sink of atmospheric vapors: the effect of vapor properties and the resulting uncertainties, Environ. Sci.: Atmos., 1, 543–557, https://doi.org/10.1039/D1EA00032B, 2021.

---

## Referee Report (RR1)

Comments on the revised version of the manuscript "Estimation of sulphuric acid concentrations…" by Lisa Johanna Beck et al.

The authors have taken up most of the comments by the reviewers and it has considerably improved. However, there is still one remaining issue that should be addressed in a minor revision, which basically concerns the general comments of the original manuscript, which are recalled in the following:

1. "The balance equations (1) to (4) are a simplification probably containing the main processes. However, also with respect to the Lovejoy et al. (2004), they do not consider several processes of impact on ambient ions, perhaps most prominent the recombination and the clustering of sulfuric acid ion clusters with water and base molecules. The effect of losses due to recombination with positive ions should be discussed. Further, the APi-TOF may not show real ambient ion clusters as in the process of pumping away neutral molecules and transfer of ions into the high vacuum TOF region, weakly bound molecules are expected to be dissociated from the clusters in collisions. And condensation sink is, as correctly stated, expected to be dependent on mass and size of the clusters. Yet, effects are expected to be minor but should be discussed.

2. The made simplifications give rise to the following issue: each budget equation, excluding eq.(1), can be solved for H2SO4 on itself. In pseudo-steady state, (2) then yields

$$[H2SO4] = CS\,[SA_{dimer}] / (k1\,[SA_{monomer}] - k2\,[SA_{dimer}])$$

And (3) yields:

$$[H2SO4] = CS\,[SA_{trimer}] / k2\,[SA_{dimer}]$$

The constant k2 can be estimated from Lovejoy et al. (2004) to be very close to k1.

Thus, together with eq. (8) of the manuscript, three equations to determine H2SO4 can be derived. Obviously, these yield different approximations of H2SO4. The differences are due to incomplete balances and the made assumptions. It is recommended and expected that the authors discuss the corresponding differences."

It is appreciated that the authors now discussed the ion-ion-recommendation, effects of clusters with water and base molecules and thus the formulations in equations (1) to (8) improved. Also it is well recognized that the authors followed the suggestion to discuss the balance equations solved for sulfuric acid based on (2) and (4) in Fig. 4b.

However, the statements on page 9 of the track-changes revised manuscript "The estimated $H_2SO_4$ concentration from Eq. 2 is highly overestimated, since the losses of the $SA_{dimer}$ to the $SA_{trimer}$ are neglected… Eq. 4 … vastly underestimating the real concentrations." Are not convincing. The loss of dimer to the trimer is considered in Eq. 2 in the term "$-k_2\,[SA_{dimer}]$ $[H_2SO_4]$". And the argument that the loss of $H_2SO_4$ due to monomer and dimer production is not considered is not correct because Eq. 4 is a budget for trimers and not for $H_2SO_4$. If Eq. 2 and 4 would describe the correct budget including all relevant processes, they should yield the same concentration of sulphuric acid each. However, they are approximations and miss some budget relevant terms, obvious from the different results for sulfuric acid in Fig. 4a and b. And, as the Eq. 2 based budget overestimates, and Eq. 4 based budget underestimates the measured sulfuric acid, Eq. 8 formed from Eq. 2 and 4 partially compensate these effects and better fit the observations.

It is recommended that the authors revise the respective statements on page 9 (ll. 278-283), and critically justify the use of Eq. 8 pointing out the best fit with observations and thus

judging the use of Eq. 8. However, associated generalisations in the conclusion should contain some appropriate carefulness.

Smaller comments (referring to lines in track-changes manuscript):

l. 180-185: Effects on CS of factor 2 are discussed here which might explain some of the deviations between the simple budgets and observations. Please comment.

l. 287: Assuming the CS at Neumayer Station to be constantly $1 \cdot 10^{-3}$ $s^{-1}$ appears to be a substantial simplification and might explain the offset between estimate and observation. Please comment.

---

## Author Response (AR2)

On behalf of the co-authors, I would like to thank the reviewer for the comments and inputs. Kindly find our responses to the comments below, which are written in green font.

Comments on the revised version of the manuscript "Estimation of sulphuric acid concentrations…" by Lisa J. Beck et al.

The authors have taken up most of the comments by the reviewers and it has considerably improved. However, there is still one remaining issue that should be addressed in a minor revision, which basically concerns the general comments of the original manuscript, which are recalled in the following:

1. "The balance equations (1) to (4) are a simplification probably containing the main processes. However, also with respect to the Lovejoy et al. (2004), they do not consider several processes of impact on ambient ions, perhaps most prominent the recombination and the clustering of sulfuric acid ion clusters with water and base molecules. The effect of losses due to recombination with positive ions should be discussed. Further, the APi-TOF may not show real ambient ion clusters as in the process of pumping away neutral molecules and transfer of ions into the high vacuum TOF region, weakly bound molecules are expected to be dissociated from the clusters in collisions. And condensation sink is, as correctly stated, expected to be dependent on mass and size of the clusters. Yet, effects are expected to be minor but should be discussed.

2. The made simplifications give rise to the following issue: each budget equation, excluding eq.(1), can be solved for H2SO4 on itself. In pseudo-steady state, (2) then yields

$[H2SO4] = CS\ [SA_{dimer}] / (k1\ [SA_{monomer}] - k2\ [SA_{dimer}])$

And (3) yields:

$[H2SO4] = CS\ [SA_{trimer}] / k2\ [SA_{dimer}]$

The constant k2 can be estimated from Lovejoy et al. (2004) to be very close to k1.

Thus, together with eq. (8) of the manuscript, three equations to determine H2SO4 can be derived. Obviously, these yield different approximations of H2SO4. The differences are due to incomplete balances and the made assumptions. It is recommended and expected that the authors discuss the corresponding differences."

It is appreciated that the authors now discussed the ion-ion-recommendation, effects of clusters with water and base molecules and thus the formulations in equations (1) to (8) improved. Also it is well recognized that the authors followed the suggestion to discuss the balance equations solved for sulfuric acid based on (2) and (4) in Fig. 4b.

However, the statements on page 9 of the track-changes revised manuscript "The estimated $H_2SO_4$ concentration from Eq. 2 is highly overestimated, since the losses of the $SA_{dimer}$ to the $SA_{trimer}$ are neglected… Eq. 4 … vastly underestimating the real concentrations." Are not convincing. The loss of dimer to the trimer is considered in Eq. 2 in the term "$-k_2\ [SA_{dimer}]\ [H_2SO_4]$". And the argument that the loss of $H_2SO_4$ due to monomer and dimer production is not considered is not correct because Eq. 4 is a budget for trimers and not for $H_2SO_4$. If Eq. 2 and 4 would describe the correct budget including all relevant processes, they should yield the same concentration of sulphuric acid each. However, they are approximations and miss some budget relevant terms, obvious from the different results for sulfuric acid in Fig. 4a and b. And, as the Eq. 2 based budget overestimates, and Eq. 4 based budget underestimates the measured sulfuric acid, Eq. 8 formed from Eq. 2 and 4 partially compensate these effects and better fit the observations.

It is recommended that the authors revise the respective statements on page 9 (ll. 278-283), and critically justify the use of Eq. 8 pointing out the best fit with observations and thus judging the use of Eq. 8. However, associated generalizations in the conclusion should contain some appropriate carefulness.

We thank the reviewer for pointing out this mistake, as this was wrongly stated in the revised manuscript. We therefore corrected the statement to the following:

*"For the sake of completeness, the estimation of the H2SO4 concentration determined from Eqs. 2 and 4, assuming pseudo-steady state, are depicted in Fig. 4b."* The estimated $H_2SO_4$ concentration from Eq. 2 is overestimating, while solving Eq. 4 for $H_2SO_4$ is underestimating the real concentration as those equations are only approximations. By combining the various approximations, Eq. 8 yields in the best fit to the observed SA concentration.

Smaller comments (referring to lines in track-changes manuscript):

l. 180-185: Effects on CS of factor 2 are discussed here which might explain some of the deviations between the simple budgets and observations. Please comment.

We added to the paragraph as follows (colored in green):
*The enhancement of CS due to the presence ions has been shown to reach a maximum value of 2 when the pre-existing particle population is centered at very small sizes (< 10 nm in diameter) and to decrease toward the value of 1 when it is located at sizes > 100 nm (Mahfouz and Donahue, 2021).* *The impact of ions on CS and estimated SA concentrations depends thereby on the environmental conditions determining the size distribution and charges of the pre-existing particle population. Neglecting the size-dependency of CS between the SA monomers, dimers and trimers causes additional errors in estimated SA concentrations; however, it is difficult to determine this effect in ambient measurements having limited data and instrumentation.*

l. 287: Assuming the CS at Neumayer Station to be constantly $1 \cdot 10^{-3} \, s^{-1}$ appears to be a substantial simplification and might explain the offset between estimate and observation. Please comment.

We agree with that and therefore added to the paragraph as follows (colored in green):

*The presented method was also applied to measurements taken at the Neumayer Station III, Antarctica, in order to test it in a different environment. Here, we used the condensation sink reported by Weller et al. (2015) at Neumayer Station of $1 \times 10^{-3} \, s^{-1}$. Figure 6 shows a three-week period between 24 December 2018 and 14 January 2019. The calibration factor of the CI-APi-TOF used for measuring the sulfuric acid concentration is $4.9 \times 10^9$. Here, the estimated sulfuric acid concentration underestimates the measured concentration when the $SA_{tetramer}$ and $NH_3(H_2SO_4)_3HSO_4$-cluster show high concentrations (Fig. 6c). A possible explanation for the underestimation might be the neglect of the growth of sulfuric acid to oligomers larger than the tetramer, as well as its clustering with bases and water (Fig. 6b and c). In coastal Antarctica, the main nucleating mechanism was observed to be negative ion-induced sulfuric acid-ammonia nucleation, acting as a major sink for sulfuric acid molecules due to its clustering with bases (Jokinen et al., 2018). Including the $SA_{tetramer}$ and $SA_{tetramer}$ clustered with $NH_3$ in the estimation equation improved the correlation ($R^2$) from 0.48 to 0.54. Furthermore, as mentioned above, the value of CS for Neumayer was*

*assumed to be constant ($10^{-3}$ $s^{-1}$) due to the lack of data needed for its calculation. This simplification certainly causes additional errors in estimated SA concentrations, especially during periods of high sea salt concentrations causing potentially large variations in values of CS.* *Nevertheless, the diurnal variation of the SA concentration is represented well by this method. During times with lower sulfuric acid concentrations, our method gives higher values than the measured concentrations (Fig. 6).*